# Mechanical breathing in organic electrochromics

Xiaokang Wang [1], Ke Chen[2], Luize Scalco de Vasconcelos [1], Jiazhi He[2], Yung C. Shin[1], Jianguo Mei [2]* & Kejie Zhao [1]*

The repetitive size change of the electrode over cycles, termed as mechanical breathing, is a crucial issue limiting the quality and lifetime of organic electrochromic devices. The mechanical deformation originates from the electron transport and ion intercalation in the redox active material. The dynamics of the state of charge induces drastic changes of the microstructure and properties of the host, and ultimately leads to structural disintegration at the interfaces. We quantify the breathing strain and the evolution of the mechanical properties of poly(3,4-propylenedioxythiophene) thin films in-situ using customized environmental nanoindentation. Upon oxidation, the film expands nearly 30% in volume, and the elastic modulus and hardness decrease by a factor of two. We perform theoretical modeling to understand thin film delamination from an indium tin oxide (ITO) current collector under cyclic load. We show that toughening the interface with roughened or silica-nanoparticle coated ITO surface significantly improves the cyclic performance.

---

[1] School of Mechanical Engineering, Purdue University, West Lafayette, IN 47907, USA. [2] Department of Chemistry, Purdue University, West Lafayette, IN 47907, USA. *email: jgmei@purdue.edu; kjzhao@purdue.edu

Organic electrochromic devices (OECDs) emerge in the avenues of smart windows[1–4] and displays[5–8] and present major advantages such as short switching time[9], multicolor capabilities[10], ambient solution processing[11], and low cost[12]. OECDs are typically composed of five stacking layers: a transparent current collector, an electrochromic layer, an electrolyte, an ion storage layer, and a second transparent/reflective counter electrode[3]. During bleaching/charging, the applied voltage drives electron extraction from the electrochromic layer (p type) with a change of its absorption band, which bleaches the polymer film[13]. Meanwhile, counterions intercalate into the electrode film to maintain electroneutrality[14,15]. The electrostatic force and mass transport collectively cause expansion in volume of the film[16]. The electrochemical process is reversed during electrochromic coloring/discharging. OECDs in practice often requires a stable cycling for hundreds of thousands of times. Over cycles, a repetitive size change of the electrochromic layer—volumetric expansion during bleaching and shrinkage during coloring, termed as mechanical breathing, persists and eventually leads to material fatigue and structural disintegration of OECDs. The interfacial incompatibility and detachment during operation[17,18] become a key factor limiting the quality and lifetime of OECDs and present an obstacle to the large-scale use of OECDs[19].

Material deformation associated with redox reactions in electrochemical systems is well studied in the past couple of decades[20–23]. However, quantification of such chemomechanical process in situ in polymer thin films remains a grand challenge, because of the softness of the organic polymers, the complexity of the chemical composition, the challenge of measurement down to the submicron scale, and the difficulty to monitor the multilayer device in a real-time operation. The reported values of volumetric strain of polypyrrole upon redox reactions have been found in the range of a few percent to a few hundreds of percent[24–26]. This huge variation comes partially from the inaccuracy of the probing technique. For instance, using the servo-controller, tensile force inevitably builds up in thin films against gravity which compromises the measurement of the actual deformation[26]. On the other end, the electrochemistry strain microscopy is sensitive to local environmental noise and might overlook the macroscopic deformation[27]. There is a need of an accurate yet facile method to detect the chemomechanical strain in redox active polymers in situ and in operando.

The change of the material state in the redox reactions often induces a mechanical breathing strain and a dynamic change of the mechanical properties of the polymers, although there is little consensus in existing studies on how the mechanical behavior quantitatively evolves over electrochromic processes. Previous measurements of the mechanical properties of poly(3,4-ethylenedioxythiophene) (PEDOT) using acoustic impedance showed that the shear modulus was sensitive to the doping level[28–30], temperature[28,29], electrolyte[28,29], crosslinker[31,32], and even film thickness[33]. Ispas et al.[28] concluded that anion insertion stiffened the PEDOT film while cation expulsion caused softening. This contradicts the recent finding by Wang and co-workers via electrochemical quartz crystal microbalance with dissipation (EQCM-D)[15] that the poly(2,2,6,6-tetramethylpiperidinyloxy-4-yl) film softens with an increase in mass while the material is in 0.5 M LiCF$_3$SO$_3$. It is worth noting that both the acoustic impedance and the EQCM-D measurements are based on presumed knowledge of the compositional fraction or the stress–strain constitutive relationship of the material. A direct method without assuming the material behavior will be advantageous to measure the mechanical properties of redox active polymers.

In the multilayer structure of OECDs, the breathing strain in the polymeric thin film is bounded by the underneath inactive substrate, typically the current collector indium tin oxide (ITO). This mismatch induces mechanical stresses in both the film electrode and the substrate. Sen et al.[34] employed multibeam optical stress sensor and showed that the stress in the polypyrrole-electrode double layer accumulated to be over 15 MPa after 50 redox cycles. The growth of the bulk stress in the organic film as well as the interfacial stress between the soft polymer and the hard substrate can cause bending of the thin double layer, wrinkling of the film electrode, crack at the interface, and debonding of the thin film from its electron conduction network. Although tremendous efforts have been placed in synthesizing new materials and modifying the interfacial adhesion[4,17,35–40], the mechanistic understanding of the damage initiation and evolution in organic thin film electrochromics remains elusive. The rational design of OECDs of enhanced mechanical reliability requires careful analysis on the generation of mechanical strain, the growth of stresses, the translation of mechanical failure into the degradation of device performance, and then a guidance of design to identify key parameters to optimize in future experiments.

Here we use the well-documented poly(3,4-propylenedioxythiophene) (PProDOT) as a model system to study the mechanical breathing strain upon the redox reactions and the failure at the interface of the device. Moreover, the methodologies and understanding can be referenced to a large library of high-performance electrochromic materials made of PProDOT. We use the environmental nanoindentation to determine the volumetric strain of PProDOT thin films during electrochromic switching in the liquid electrolyte and then to measure the mechanical properties in the reduced and oxidized states. The thin film electrode expands up to 30% in volume upon oxidation and both elastic modulus and hardness decrease by a factor of two. We then perform computational modeling to examine the stress field and interfacial failure between an ITO current collector and the film. The stress concentration initiates an edge crack, which continuously enlarges toward the center of the film driven by the shear cracking during oxidation and a mixed mode of shearing and opening crack during reduction. The damage evolution is in excellent agreement with our in situ observation. We use the dimensionless quantities of the breathing strain and the crack driving force to draw a phase diagram to delineate the safe and delamination zones. To demonstrate the design principle of the electrochromic electrode, we show that toughening the interface with roughened or silica-nanoparticle coated ITO surface increases the cyclic life of PProDOT films by nearly two orders of magnitude.

## Results

**Mechanical behavior of PProDOT upon electrochromic reactions.** We sketch the interfacial delamination in organic thin film electrochromic devices and its impact on the cyclic performance in Fig. 1a. The mechanical debonding of the film electrode from the current collector limits electron and counterion transport and impedes electrochromic switch of the film upon cycles—at the oxidized state, the delaminated regimes retain positive charges and counterions and therefore remain their bleaching state in the following reduction reaction, while the intact regimes maintain electron and counterion transport and enable chromic switching. Supplementary Movie 1 shows the mechanical breathing and interfacial delamination of a PProDOT film after around 160 cycles by in situ optical observation. The repetitive deformation and the partial debonding of the film are visible by bare eyes. The optical microscope is located within a glovebox filled with argon. The inert environment avoids contamination of moisture and oxygen to the liquid electrolyte. Figure 1b shows a few snapshots

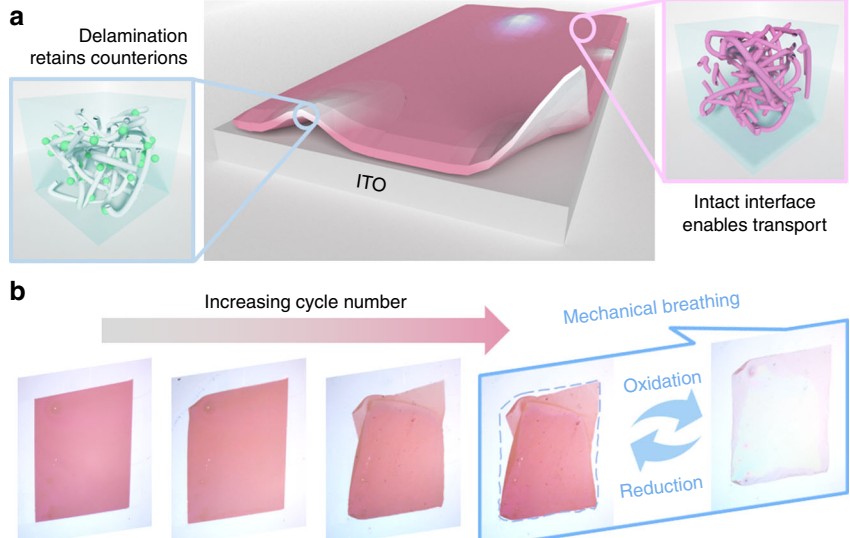

**Fig. 1 Sketch of interfacial delamination in thin film electrochromic devices. a** The mechanical delamination from the current collector limits electron and counterion transport and impedes chromic switch of the film upon electrochemical cycles. **b** Experimental observation of mechanical breathing of PProDOT film and its morphology after every 60 cycles. The thin film experiences repetitive expansion and shrinkage in volume in the redox reactions which ultimately leads to the failure of the device at the interface.

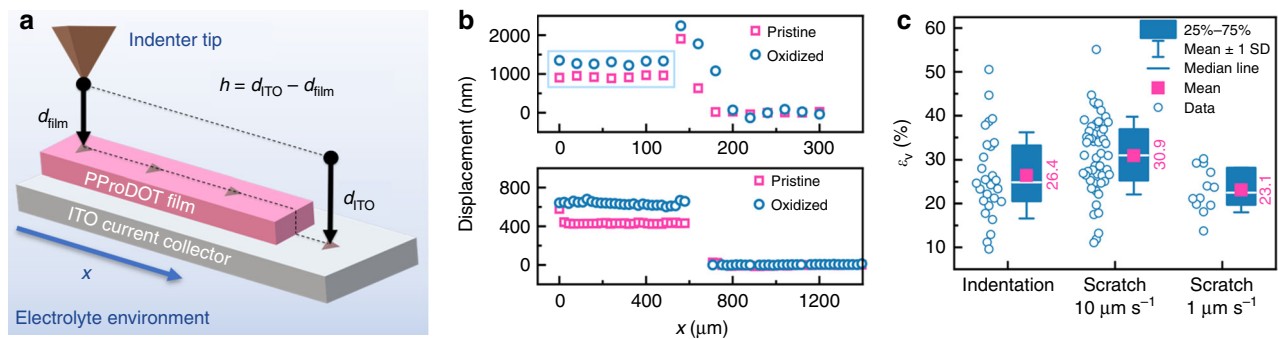

**Fig. 2 In situ thickness measurement. a** Sketch of the thin film thickness measurement by the environmental nanoindentation method. $d_{film}$ ($d_{ITO}$) denotes the travel displacement of the tip when the contact between the tip and the film (ITO) is detected. **b** Thickness of PProDOT in the pristine and oxidized states. The upper panel shows tip displacement measured by targeted indentation. The lower panel shows the tip displacement in the $x$-direction measured by the scratch test. For both methods, the step height denotes the thickness of the film. **c** A volumetric strain $\varepsilon_V$ in the range of 20–30% is determined for PProDOT upon oxidation using the scratch and targeted indentation methods.

of the film morphology after every 60 cycles starting from its pristine state. The repetitive change in size of the PProDOT electrode upon redox reactions eventually leads to the failure of the electrode at the interface.

We use customized nanoindentation to measure the breathing strain in PProDOT film on ITO via targeted indentation and scratch test. Figure 2a shows the schematic of the methodology, where $d_{film}$ denotes the travel displacement of the tip when the contact between the tip and the film is detected, and $d_{ITO}$ represents the tip displacement down to the ITO substrate. To eliminate the effect of the liquid flow, all the electrodes are firmly attached to a homemade fluid cell[41]. The abrupt change in the contact stiffness when the tip approaches to the surface, Supplementary Fig. 1, indicates the surface contact. For the targeted indentation, we sample a series of indentation points across the boundary between the film and the substrate. Possible effect of sample tilting is leveraged. The tip displacement $d_{film}$ or $d_{ITO}$ for each targeted indentation is recorded and is shown in the upper panel of Fig. 2b. The step height represents the thickness of the film. This method eliminates penetration of the tip into the sample, as is occasionally observed in the scratch

test. For this non-standard method, we use atomic force microscope (AFM) to validate the targeted nanoindentation for the dry sample. The AFM images (Supplementary Fig. 2) are taken at the same locations where indentation tests are performed. The thicknesses measured by the two methods are listed in Supplementary Table 1. The good agreement demonstrates that the targeted indentation measurement is reliable. Another independent measurement is done by the scratch test. We use the embedded G-Series Ramp Load Scratch with Topography Compensation method in the instrument. The tip profiles the surface with a very small load (1–3 µN) along a straight line crossing the boundary between the film and the substrate. The tip displacement versus the scratch distance is shown in the lower panel of Fig. 2b. Again, the step height gives the thickness of the film. Note that, the film pile-up near the boundary between the film and the substrate may bring in artifacts in the measurement and the tip may end up crashing on the film from the side. The local surface detection in this case is not accurate. Here we only use the data marked in the cyan box in the case of targeted indentation, away from the boundary, to interpret the film thickness.

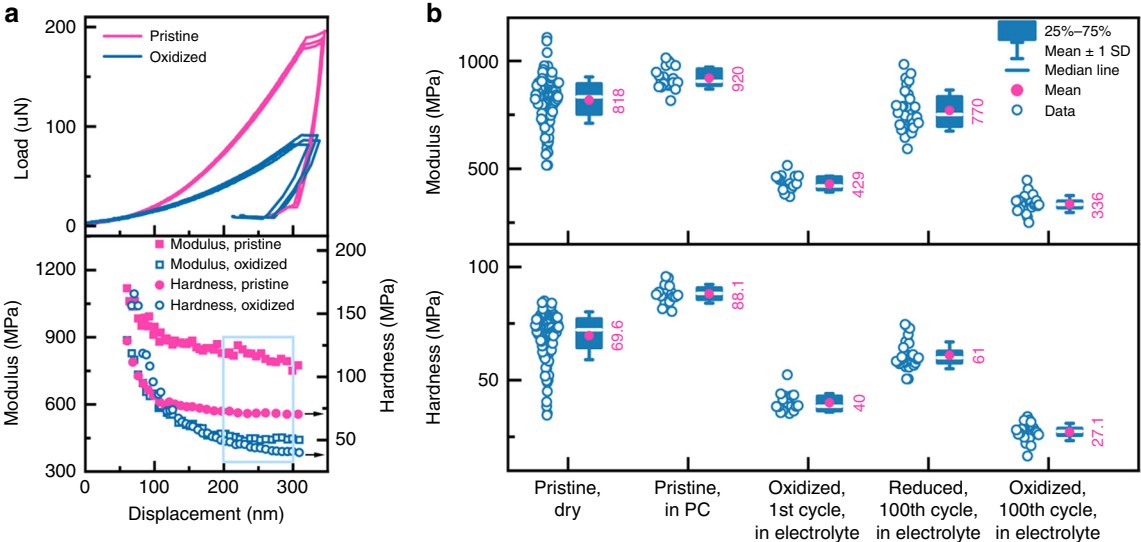

**Fig. 3 Mechanical properties of PProDOT film. a** Upper panel: the load-displacement curves of indentation on the pristine and oxidized PProDOT films. Lower panel: Modulus and hardness of the pristine and oxidized PProDOT as a function of the indentation depth. **b** Modulus (upper panel) and hardness (lower panel) of PProDOT in the pristine and dry state, the pristine state in PC, the oxidized state in electrolyte after the 1st cycle, the reduced state in electrolyte after 100 cycles, and the oxidized state in electrolyte after 100 cycles.

With the two methods described above, we measure the thickness of the film at the same locations in the pristine and oxidized states in the first cycle. The nanoindentation sites are chosen ~50 um away from the edge to avoid possible interference of film delamination from the substrate. As seen in Fig. 2b, the film surface is clearly elevated upon oxidation indicating an increase of the film thickness. For each measured location, we compare the thickness of the film before and after oxidation, $(h_0, h)$. Since the in-plane deformation of the film is bounded by the hard substrate, the volumetric strain is calculated by $\varepsilon_V = (h - h_0)/h_0$. Figure 2c shows the results of the measured volumetric strain with the average and standard deviation, the median, and the 25–75% range of the data. The volumetric strain is found to be 26.4% by targeted indentation, 30.9% and 23.1% by scratch test for the tip profiling velocity of $10\,\mu m\,s^{-1}$ and $1\,\mu m\,s^{-1}$, respectively. This overall volumetric strain gives a roughly 10% linear strain for a homogeneous and isotropic material. The deformation is recoverable if the strain is within the elastic limit and if the induced stress does not exceed the material yield strength[42]. From a microscopic perspective, the polymer chains are entangled in nature. A tensile stress elongates the bulk polymeric material by stretching the serpentine chains to a straight configuration followed by interchain slip. The intrachain elongation is often recoverable upon removal of the external load, while the intrachain slip manifests as the permanent deformation[43,44]. For the PProDOT film we study here, the averaged molecular weight $M_n = 9900$ suggests that the molecular chain is made of ~30 monomers, which corresponds to an end-to-end length of less than 10 nm. It is most likely that the interchain slip accommodates the large volumetric change of the film in the redox reaction rather than the intrachain elongation. This fact indicates that the plastic flow is invoked upon oxidation when the counterions and solvent molecules insert into the film. In the course of reduction, the counterions and solvent molecules are expelled from the host and the polymer coils aggregate by the interchain interactions.

We perform nanoindentation to measure the elastic modulus and hardness of the PProDOT film in the pristine state (dry and in PC, respectively), reduced state (in electrolyte), and oxidized state (in electrolyte) using the continuous stiffness measurement

(CSM). The load–displacement response is shown in the upper panel of Fig. 3a. A harmonic oscillation of 2 nm at 45 Hz is superposed during loading, such that the modulus and hardness can be determined as a continuous function of the indentation depth. We have eliminated the substrate effect using the prior-established model[45]. The lower panel of Fig. 3a shows that the modulus and hardness decrease as the indentation depth increases. This behavior is typical for a soft film on a hard substrate and is consistent with several prior studies[46–48]. Here we use the data in the plateau region marked in the cyan box to determine the average value. We measure the material properties of the pristine sample in both dry and wet states (in propylene carbonate for 2 h) to eliminate the potential effect of the liquid environment. The results of the pristine sample are consistent but the measurement in the liquid environment seems less spread, Fig. 3b. The same procedure is employed to determine the elastic modulus and hardness of the film after oxidation. It is striking that both the modulus and hardness decrease by nearly a factor of two when the film is oxidized as shown in Fig. 3b. And the electrochemical conditioning process, by comparing the data of the 1st and 100th cycles, has limited effect on the mechanical properties. This drastic decrease in the mechanical properties might be counterintuitive. The mechanical response is related to the change of the state of charge and the microstructural feature of the polymer chains. Upon oxidation, the neutral chains lose electrons and morph into a quinoid structure[49]. A stiffer backbone is then expected due to the nature of quinoid structure upon charge delocalization. The experimental results indicate that (1) the intermolecular interaction is mostly responsible for the mechanical response of the film, and (2) the intercalation of counterions and solvent molecules weakens the intermolecular interactions among the loosely entangled polymer chains. We are pursuing atomistic modeling to examine the local bonding environment, the long-range intermolecular interactions, and microstructural evolution to fully understand the polymeric material behavior upon redox reactions.

**Mechanistic understanding of electrochromic film delamination.** With the experimental input of the breathing strain and the

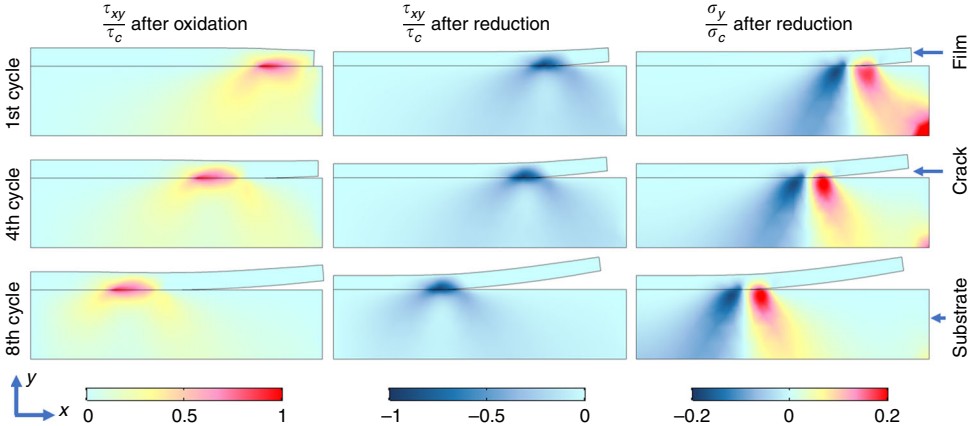

**Fig. 4** Contour plots of the shear stress $\tau_{xy}$ in PProDOT at the oxidized state, $\tau_{xy}$ at the reduced state, and the normal stress $\sigma_y$ at the reduced state after the 1st, 4th, and 8th cycles, respectively. Deformation is drawn in the actual scale.

mechanical properties of PProDOT, we conduct finite element analysis (FEA) to understand the stress field and the crack initiation and growth in the organic thin film electrochromic devices. We use an elastic-perfectly plastic constitutive relationship to describe the PProDOT film. The elastic modulus is taken from the experimental results in Fig. 3, and the material yield strength is assumed to be one third of the hardness. To mimic the volumetric expansion upon oxidation, an isotropic thermal strain up to 10% is applied to deform the film. The polymeric film expands against the constraint provided by the substrate. The interaction of the film–substrate system at the interface is described by a traction-separation law of a trapezoidal shape (Supplementary Fig. 3). When the contacting points start to separate, the interfacial traction increases linearly with a stiffness $K_i$ until it reaches the traction limit $T_{ic}$. Here $i$ denotes the normal ($i = I$) or tangential ($i = II$) load. The damage function $D$ remains 0 within the elastic regime and starts growing when $T = T_{ic}$. Following the elastic load, the interfacial traction maintains a constant to mimic the plastic flow of the film. When the dissipated energy $G_{ic}$ (shaded area in Supplementary Fig. 3) is equal to the interfacial toughness $\Gamma$, the traction reduces to 0 and the interface is fully separated ($D = 1$). Detailed description of the traction-separation law can be found in Supplementary Note 1.

FEA results show that oxidation of the film leads to the concentration of shear stress around the free edge between film and the substrate, as shown in the contour plot, left column of Fig. 4. Once the shear stress exceeds the interfacial strength, the interfacial damage initiates and grows, as is evident in the correlation between the damage function and the shear stress distribution in Supplementary Fig. 4a. The different lines in Supplementary Fig. 4a represent the various degrees of oxidation with $\varepsilon = 0.1$ representing the complete oxidation. When the oxidation reaction proceeds, the film continues to expand with a steady growth of the interfacial crack. The normal stress associated with the oxidation reaction remains compressive, therefore the damage is driven by a pure shearing crack (mode-II). In the following reduction reaction, the PProDOT film shrinks in volume against the interfacial adhesion. The stress field within the film starts to change with an elastic unloading and succeeds by an opposite shear stress and a positive normal stress. The positive normal stress is a result of the plastic flow of the film[50]. Supplementary Fig. 4b shows the evolving shear stress at the interface in an oxidation and reduction cycle. The contour plots of the shear stress and normal stress in different cycles are shown in the middle and right columns of Fig. 4. In the process of the reduction reaction, the interfacial damage is driven by a

mixed mode of shearing and opening cracks. The positive out-of-plane normal stress is the reason to cause the bending of the film and delamination from the substrate. From the computational results we understand the dynamics of the interfacial damage as follows: the breathing strain induces a mismatch strain in the film and the substrate; the constraint of the substrate causes concentration of stresses at the free edge; edge damage emerges as the stress exceeds the interfacial strength; the edge crack continuously grows toward the center of the film driven by shearing crack during oxidation and a mixed mode of shearing and opening crack upon reduction. The damage evolution in the finite element modeling agrees very well with the in situ optical observation as shown in Fig. 1b and in Supplementary Movie 1.

To draw the complete portrait of the interfacial damage in the electrochromic electrodes, we examine more closely the dynamics of the damage initiation, crack opening, and propagation. In the early stage of cycle, the interface remains intact for the regime away from the free edge. As the redox reaction proceeds, the stress in the delaminated zones is released, and the stress concentration and mechanical damage are progressively translated toward the center of the film. The intact area, the damage zone where the film and the substrate are partially separated, and the cracked regime are outlined in the upper panel of Fig. 5a. The figure also shows the shear stress profile and the interfacial damage function along the interface after the 4th oxidation reaction. The lower panel of Fig. 5a plots the size of the damage zone and the size of the crack length, normalized by the initial film thickness, as a function of the cycle number. The crack opening is an irreversible process. We observe that the size of the damage zone reaches a nearly constant value after the initial oxidation reaction albeit the stress field alternates quite dynamically afterwards. The size of the cracked zone, on the other end, increases almost linearly starting from the first reduction reaction up to the 8th cycle. This is understood due to the combination of the reversible breathing strain in the redox reactions and the plastic deformation of the film—the collective factors result in pretty much the same magnitude of the stress field except the difference in the sign of the stresses in the oxidation and reduction processes. In addition, the shear stress generated at the interface is a dominating factor driving the film delamination. Therefore, the cracked regime increases linearly in size, separated by a nearly constant damaged zone from the intact area, over cycles.

We construct a phase diagram to guide the design of the thin film electrochromic devices of enhanced mechanical reliability. By intuition, the mechanical damage depends on the breathing strain

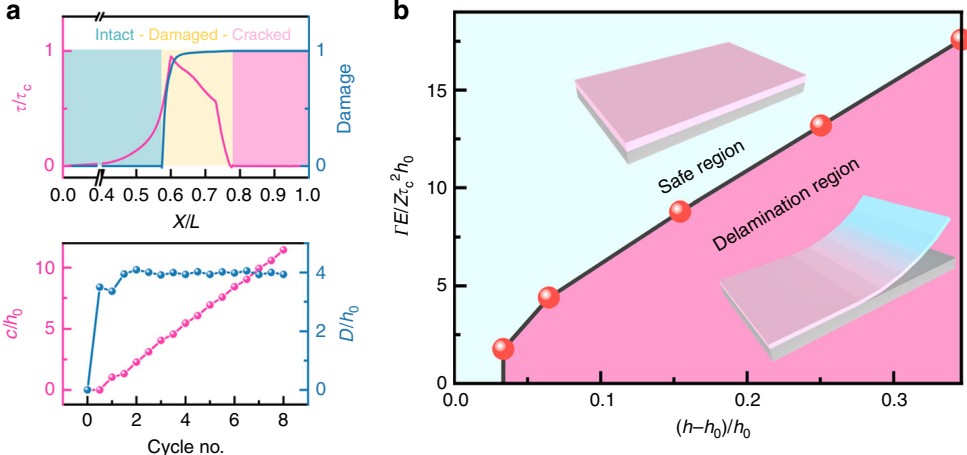

**Fig. 5 Damage analysis of PProDOT thin film upon redox reactions. a** Upper panel: Sketch of the shear stress profile (left $y$ axis) and the interfacial damage function (right $y$ axis) along the interface after the 4th oxidation reaction. Lower Panel: The evolution of the crack length (magenta dots), $c/h_0$, and the size of the damaged zone (blue dots), $D/h_0$, as a function of the cyclic number of the redox reaction. **b** A phase diagram of interfacial delamination in electrochromic thin film in the space of the dimensionless breathing strain and interfacial toughness. The solid spheres represent the numerical results, while the line is drawn to delineate the boundary between the intact and delaminated conditions.

$\varepsilon_V = (h - h_0)/h_0$ for a thin film bounded by a substrate. Crack initiates at the interface when the driving force, the energy release rate, exceeds the interfacial toughness. The energy release rate for a thin film subject to the shear yielding is calculated as $G = Z \cdot \frac{\tau_c}{E} \cdot \tau_c \cdot h_0$, where $Z$ is a dimensionless parameter describing the geometric effect, $\tau_c$ is the shear yield strength, $E$ is the elastic modulus, and $h_0$ is the film thickness. For the initiation of debonding of thin films, $Z = 1.026$[51]. The dimensionless parameter, $\frac{\Gamma E}{Z\tau_c^2 h_0}$, the interfacial toughness $\Gamma$ normalized by the energy release rate $G$, describes the competition between the crack driving force and the crack resistance. Figure 5b shows the computational results of the critical conditions to cause film delamination in terms of the dimensionless breathing strain $\varepsilon_V = (h - h_0)/h_0$ and the material parameters $\frac{\Gamma E}{Z\tau_c^2 h_0}$. The solid spheres represent the numerical results, while the line is drawn to delineate the boundary between the intact and delaminated conditions. The phase diagram offers design rules to maintain the structural integrity of the thin film electrochromic devices. Interfacial damage will less likely happen by (1) minimizing the breathing strain in the redox active thin films, (2) enhancing interfacial toughness $\Gamma$, (3) utilizing materials of a higher elastic modulus $E$ and a lower yield strength $\tau_c$, and (4) reducing the film thickness $h_0$. In short, the general guideline is to use small-size, stiff (high modulus), and soft (low yield strength) film electrode, and tough interfacial adhesion.

**Interfacial engineering for enhanced mechanical reliability.** For the fabrication and device performance, the thickness of the film electrode is often chosen to maximize the optical contrast between the two redox states. Among the rules offered by the phase diagram, the interfacial toughening by physical or chemical modification seems most practical. While providing enhanced adhesion, the modified interface should be highly transmissive, have good electron-transport properties and remain of low cost. Current strategies include chemical bonding, physical bonding, and surface roughening to enable mechanical interlock of the film and the substrate. Here we demonstrate that by grinding the pristine ITO surface using sandpaper and by coating the interface between the current collector and the polymeric film with silica-nanoparticles (SiO$_2$ NP), the cyclic life (when current density > 0.15 mA cm$^{-2}$, Fig. 6f–h) of the electrochromic electrode is

promoted considerably as compared to the bare ITO electrode (by nearly two orders of magnitudes for SiO$_2$ NP-treated ITO), Fig. 6a–e.

The PProDOT thin film electrodes start from the same condition (morphology and interfacial conductivity), as indicated from the pristine state of electrodes (Fig. 6b) and similarly among the first-three cyclic voltammograms (CVs) cycles in both shape and the current density (Fig. 6f–h). The CVs of PProDOT thin films on both the bare ITO and modified ITOs show a pair of redox peaks at 0.56 V and 0.29 V and the same onset of the oxidation potential of ~0.4 V, which indicates that the surface modifications have negligible effects on the electrochemical characteristics of PProDOT thin films. The current density for all three electrodes gradually drops in subsequent cycles, possibly due to micro-scale delamination and ion trapping[52] untill obvious film delamination is observed. The PProDOT film on bare ITO is severely damaged after 140 cycles (Fig. 6c), leaving only the magenta part in contact while the remaining region being peeled off and dysfunctional. The charge density quickly drops from 4.87 mC cm$^{-2}$ to 1.8 mC cm$^{-2}$. Part of the PProDOT film on roughened ITO is delaminated after 380 cycles and reaches the same electron density of 1.8 mC cm$^{-2}$ from 4.75 mC cm$^{-2}$ (Fig. 6d). In contrast, the PProDOT film on SiO$_2$ NP treated ITO starts with an electron density of 4.62 mC cm$^{-2}$ and sustains over 8500 cycles before its current density decreases to the same value of 1.8 mC cm$^{-2}$. In this case, it seems that only minor delamination occurs as observed at the end of cycles (Fig. 6e). The interface damage is also evident by the drop in the current density as shown in Fig. 6f, g.

The improved durability is attributed mostly to the increased surface roughness of the ITO which enables mechanical interlock and reinforces the adhesion of the film[53]. This is demonstrated by the surface morphology and roughness of the bare ITO, ITO ground by sandpaper, and SiO$_2$ NP coated ITO in Fig. 7 and Supplementary Fig. 5. The bare ITO has the finest surface with a root mean square height of only 5.51 nm, followed by SiO$_2$ NP treated ITO surface (21.9 nm). The nanoparticles (diameter of ~200 nm) self-assemble into a well-packed hierarchical nanostructure, as shown in the 3D AFM image in Fig. 7d and in the SEM images in Supplementary Fig. 6. Nanoscale inter-particle gaps (white arrow in Supplementary Fig. 6b) introduces high-density mechanical interlock between the polymer film and the substrate, which significantly improves the performance. Note

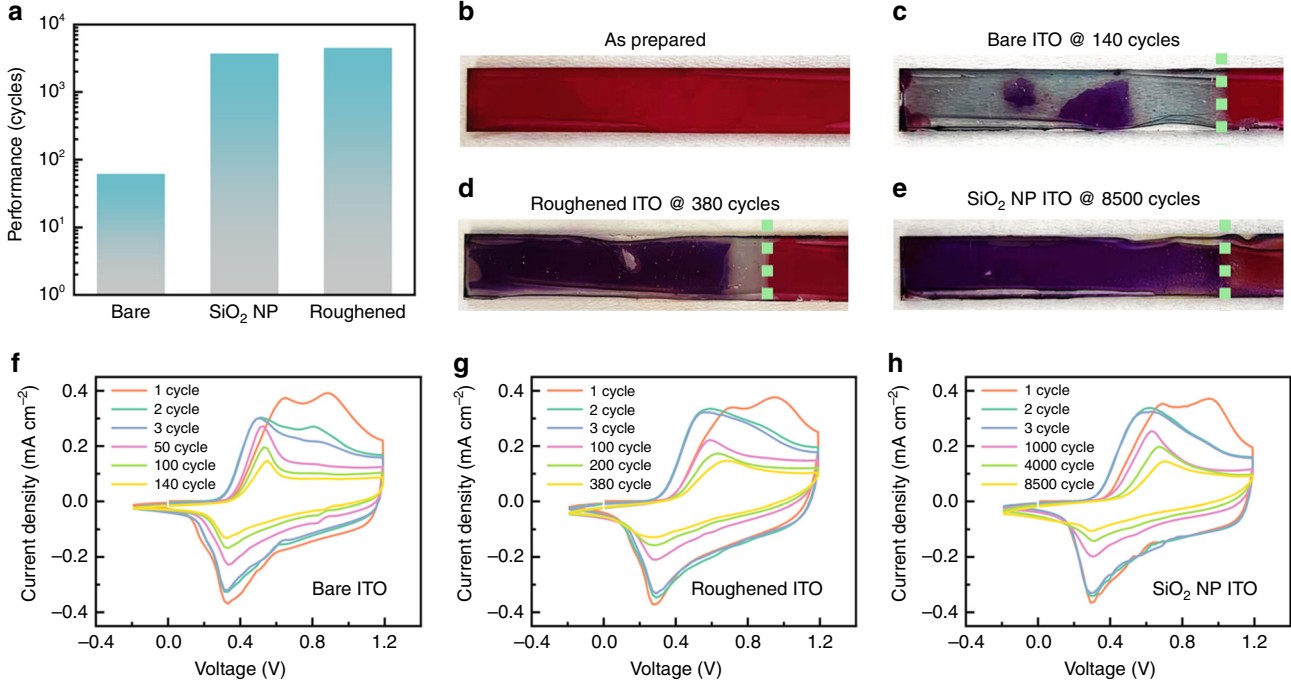

**Fig. 6 Interfacial modification of electrochromic electrode. a** The surface treatment and improvement of interfacial contact considerably enhance the cyclic performance of OECDs. (**b**)-(**e**) show the images of the as-prepared PProDOT film on bare ITO, PProDOT on bare ITO after 140 cycles, PProDOT on roughened ITO after 380 cycles, and PProDOT on SiO$_2$ NP treated ITO after 8500 cycles, respectively. The cyan dot lines indicate the electrolyte front line. (**f**)-(**h**) show the cyclic voltammetry responses of PProDOT film on bare ITO, roughened ITO, and SiO$_2$ NP-treated ITO, respectively.

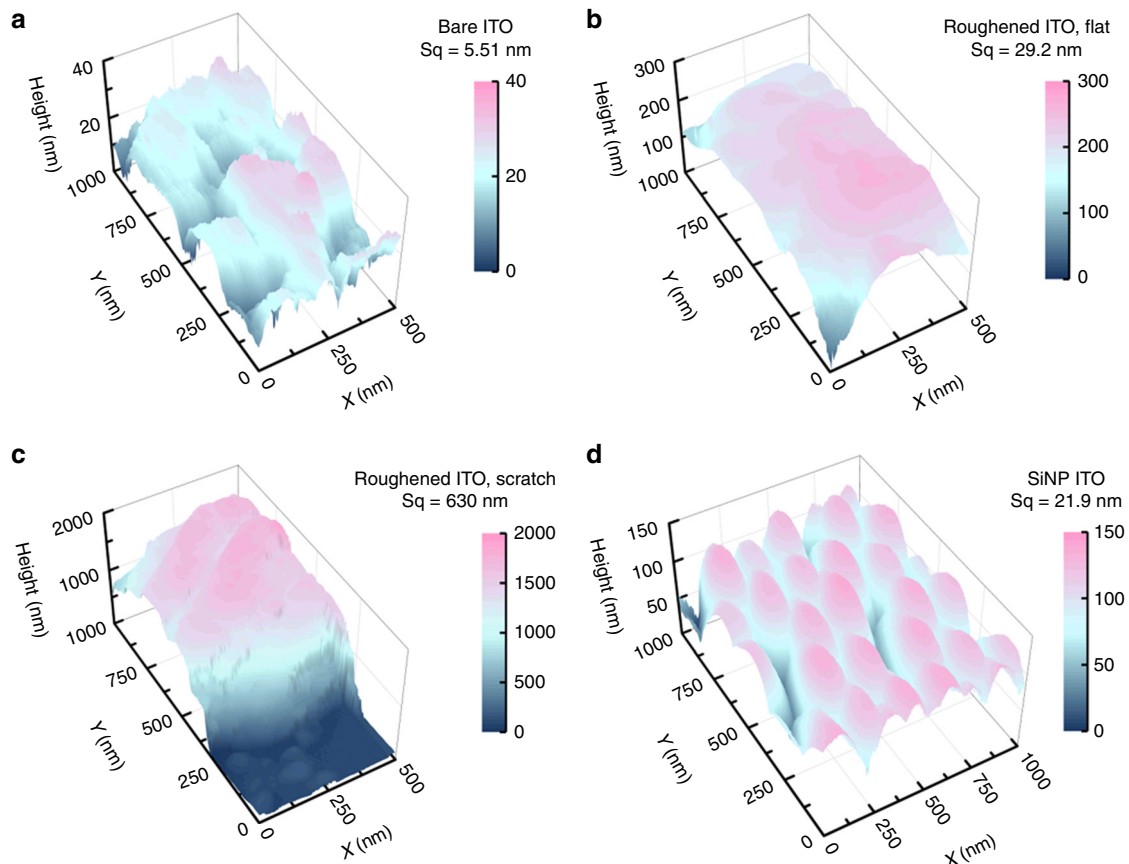

**Fig. 7 3D surface morphology of the ITO surface. a** Bare ITO. **b** A flat region in roughened ITO. **c** A scratched region in roughened ITO. **d** SiO$_2$ NP-treated ITO. Sq denotes the root mean square height roughness.

that the mud cracks (red arrows) in Supplementary Fig. 6a and 6b are formed by the electron-wind forces at high magnification and are absent from the modified electrodes. Due to the size of the abrasion particle on sandpaper, the roughness of the ground ITO surface varies from 29.2 nm to 620 nm. The characteristic size in ground ITO electrode is in the micron scale, rendering a less dense mechanical interlock and less improved cyclic life. In addition to the surface roughness, $SiO_2$ NP can also change the physical properties of the ITO surface which benefits the interfacial adhesion[35,54].

## Discussion

In summary, we employ customized environmental nanoindentation to probe the breathing strain of electrochromic thin films in situ upon cyclic redox reactions. The PProDOT film deforms up to 30% in volume in the oxidation and reduction processes. The variation of the state of charge alters the elastic modulus and hardness by a factor of two and the film becomes softer and more compliant in the oxidized state. We perform theoretical modeling to understand the damage initiation and propagation at the interface of electrochromic layer and the current collector. The mechanical breathing of the redox active film induces major stresses near the free edge between the film and the substrate. Edge crack emerges when the mismatch stress exceeds the interfacial strength. The oscillatory load, resulted from the repetitive size change of the film in the redox reactions, alters the stress field, and leads to a linear progression of film delamination toward the center over cycles. The breathing strain in the electrochromic film and the dynamics of the interfacial damage are in excellent agreement with the in situ optical observation. We construct a phase diagram, in terms of the dimensionless quantities of the breathing strain and the material parameters, to guide the design of the thin film electrochromic devices of optimum mechanical stability. We demonstrate the design rules by toughening the interface with roughened or silica-nanoparticle coated surface which results in an elongated cycle lifetime of nearly two orders of magnitude compared to the untreated sample.

## Methods

**Film processing**. The PProDOT was synthesized via direct arylation polymerization using a previously reported procedure[55]. The molecular weight was characterized by gel permeation chromatography. Then PProDOT was dissolved in chloroform and stirred overnight to form homogeneous solution with concentration of 40 mg mL$^{-1}$ before use. ITO-coated glass slides were cleaned with ultrasonic successively in chloroform and ethanol for 10 minutes. The PProDOT solution was then spin coated on ITO-coated glass slides with a spin speed of 800 rpm and 600 rpm to generate films of thicknesses of ~500 nm and ~1000 nm, respectively.

**Surface modification of ITO**. For surface roughening, two modification methods were employed to increase the roughness of ITO surface. In the first method, the ITO was ground by P1200 Starcke silicon carbide sandpaper. Very gentle force was applied in two orthogonal directions in sequence to generate visible clouds on ITO surface. The ground ITO was then cleaned through the processes described in the film processing part. A second method of modification involves the coating of $SiO_2$ nanoparticles after cleaning. Monodisperse $SiO_2$ nanoparticles with the diameter around 200 nm were synthesized by Stöber method[56] and then were dispersed in EtOH by sonication to form homogenous solution with concentration of 0.13 g mL$^{-1}$. The solution was then spin coated on the pre-cleaned ITO/glass with spin speed of 1500 rpm. After being put in the 90 °C oven for a few minutes, the EtOH volatized completely, which produced a solid $SiO_2$ film on the ITO/glass substrate. The control experiment was done using as-received ITO after the same cleaning procedure.

**Electrochemistry test**. To allow indentation on the thin film, a half-cell configuration was used. The PProDOT film on ITO, a Pt wire, and a homemade Ag/AgCl wire[57] were used as the working electrode, counter electrode, and reference electrode, respectively. 1 M LiPF$_6$ in propylene carbonate (PC, Sigma Aldrich) was used as the electrolyte. All the electrochemistry experiments were done at the Versa Stat

3 potential station. For indentation test and scratch test on oxidized films, a voltage of 1 V against the reference electrode was applied. For the durability test, a three-electrode cell was fabricated for the cyclic test with 0.2 M LiTFISI in PC as the electrolyte. Voltammetry experiments were performed between 1.2 V and −0.2 V with a scan rate of 40 mV/s. The charge density was calculated by the equation $\int \frac{jdV}{s}$, where the charge density has the unit of mC cm$^{-2}$, $j$ is the current density (mA cm$^{-2}$), $s$ is the scan rate (V s$^{-1}$), and $V$ is the voltage (V).

**Targetted indentation and scratch tests**. Instrumented nanoindentation (G200 Nano-indenter, KLA) was implemented to probe the mechanical properties of the films. All tests were done in an Ar filled glovebox to eliminate the chemical degradation by moisture and oxygen. During the test, load–displacement curve was recorded, from which modulus and hardness were calculated. The thin film method was used to calculate modulus, as implemented in the NanoSuite software[45]. To measure the thickness of the films, the raw displacement method and scratch test were used. For both methods, the indenter tip approaches the film until the surface is found. The recorded raw displacement at detected surface unveils the thickness of the film (Fig. 2a). Detailed explanation can be found in the Results section. The AFM (Veeco Multimode) data for comparison was processed via an online tool at https://www.profilmonline.com/.

**Finite element analysis**. To explore the degradation mechanism during the cyclic redox reactions, FEA was implemented using COMSOL Multiphysics $^{TM}$ (COMSOL Multiphysics 5.3, Sweden). A soft, compliant thin film (thickness of 500 nm, width of 10 um) is on a hard, stiff substrate. Elastic and perfectly plastic relationship was assumed for the polymer film. The modulus, 809 MPa, was measured from the indentation test. The yield stress, 23.2 MPa, was estimated to be 1/3 of the hardness[58]. The substrate deforms elastically with a modulus of 80 GPa. The interface debonding was simulated by the cohesive zone model[59]. Maximum normal (shear) strength was set as $\sigma_y$ ($\sigma_y/\sqrt{6}$) such that the interfacial opening crack (sliding) occured upon yielding of the film. Interfacial fracture toughness was estimated to be 1 J m$^{-2}$ for both mode I and mode II cracks[60]. Analogous to the thermal expansion, an isotropic strain of 10% was applied to the film upon oxidation and then reduced to 0 during reduction. The effect of the mesh size was tested and convergence of results was obtained.

## Data availability

The data that support the findings of this study are available from the corresponding author upon reasonable request.

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

## Acknowledgements

K. Z. acknowledges the support by the National Science Foundation through the grant CMMI-1726392. J. M. is grateful for the support from Ambilight Inc.

## Author contributions

X. W., K. Z. and J. M. conceived the idea and designed the in situ experiments. X. W. and L. V. performed the in situ measurements. X. W. performed the finite element analysis. K. C. and H. J. performed the interfacial modification of the electrode and the electrochemistry characterization. X. W. conducted optical profilometer measurement under the guidance of Y. S. All authors contributed in writing the paper.

## Competing interests

The authors declare no competing interests.
