## [Peer Review File · Nature Communications]

Reviewers' comments:

Reviewer #1 (Remarks to the Author):

This contribution details the factors that contribute to mechanical delamination of electrochromic polymers from ITO during cycling. The authors employ nanoindentation and modeling of a promising and well-studied polymer, PProDOT. I think the work is generally of interest but I have concerns over data interpretation (comment 9) and the use of SiO₂ (comment 16). Parts of the paper are a bit hard to get through and easily read.

2. Page 3 Paragraph beginning with "In concomitant..." The authors try to generalize behavior of the mechanical properties of electrochemically active polymers. This is dangerous because it is related to the ion type, solvent type, and polymer type. This makes comparisons across refs 28-34 difficult because they have different ions, solvents, and polymers.

3. It is my view that the solvent is very important to the mechanical properties, and how the solvent transports in and out of the film, accompanied by an ion's solvation shell etc.

4. Page 3, "Previous measurement of..." The way it is written, it implies that refs 28-34 all study PEDOT, which is unclear.

5. Page 8. The equation for volumetric strain assumes a constant area, but the video clearly shows changes in area. Authors should discuss.

6. "This overall volumetric stain gives a roughly 10% linear strain for a homogeneous and isotropic material." This sentence is confusing given comment #6.

7. Figure 3 and caption should be understandable as a self-contained unit. Panel (a) – is it dry or in PC? Panel (b) – are "In PC" samples reduced or oxidized. Panel (b) are "Oxidized" samples dry or in PC?

8. "We perform nanoindentation to measure the elastic modulus and hardness of the PProDOT film in the pristine and oxidized states using the continuous stiffness measurement 1 (CSM)." Dry or in PC?

9. I have concerns with Figure 3. Many electrochemically active polymers need conditioning for electrolyte penetration. I am not sure that the authors considered this. I think that the "In PC" and "Oxidized" measurements are just a result of the electrode not being properly conditioned. To resolve this the authors should present data on "Uncycled In PC", "Conditioned and Reduced", and "Conditioned and Oxidized". Then it will be a clearer result. Without addressing conditioning, I fear their discussions on pages 9-10 are misguided.

10. Figure 4 is hard to interpret for the non-expert. The ITO and polymer film should be labeled. The crack propagation or delamination should be indicated, perhaps with an arrow.

11. Page 15 "In short, the general guideline is to use small-size, stiff, and soft film electrode, and tough interfacial adhesion." How can an electrode be both "stiff" and "soft"? That seems contradictory. Same issue in the conclusion section.

12. Page 14. The energy release rate G depends on modulus and yield strength, but these vary between oxidized and reduced states. How did the authors account for this? Or what assumptions went into the calculation?

13. "For a given selection of the polymeric film, the mechanical properties are about fixed." Confusing. What is really meant here?

14. Please do not use SiO₂ as (SiNP) because in the silicon anode community that term means "Si" pure nanoparticles – not SiO₂, so it will confuse audience.

15. Figure 6. Do all electrodes have the same thicknesses?

16. Why chose SiO₂? It is insulating and not an obvious choice. I am surprised that an SiO₂ interface still allows for electrochemical activity between the ITO and the polymer.

Reviewer #2 (Remarks to the Author):

The paper describes a very informative study on one of the most relevant topics in organic electrochromic devices: the lack of stability after repeated cycling due to delamination. OEC materials have advantages over the corresponding inorganic counterparts such as WO₃, yet the relatively poor stability they feature because of intrinsic chemical sensitivity to oxygen and water and mechanical instability in multilayer devices so far hampered the commercial valorisation of results.

If careful (and expensive) encapsulation can dramatically improve chemical sensitivity, the problem of delamination and cracking connected with the large volume variation upon oxidation/reduction is more difficult to tackle.

I particularly appreciate the results here described because the quantitative evaluation of the extension of volume variation is professionally carried out leading to values that are fully believable, and very useful for the community.

Also, to the best of my knowledge this is the first time that the changes in the mechanical properties upon doping are directly measured, and with very surprising results. I did not expect the ProDOT to soften upon chemical doping, yet the data are believable and the interpretation is sound.

The finite element analysis of the crack propagation is also very relevant, particularly as useful guidelines can be obtained in order to design improved working electrodes (essentially by improving adhesion).

I have only one relevant concern about the paper: generality.

Essentially all aspects here described are bound to be solvent and electrolyte dependent. Indeed, the stability of ProDOT upon cycling was found to be very good already quite a few years ago, when working with acetonitrile as the solvent. (10.1002/adma.200300376). Multiple reports highlight both the high electrochromic contrast and high stability of such a polymer.

I do not doubt the data described here but it looks like the particular solvent here employed was selected to highlight the degradation by delamination phenomenon. I was particularly surprised by the data shown in figure 6c of such a severe damage after just 140 cycles.

I strongly suggest to repeat the stability upon cycling test in at least another solvent (acetonitrile could be a good choice) and to use an electrolyte more relevant for devices (LiTFSI could be good).

Again on the same topic, in solid state devices the liquid electrolyte is generally replaced by a gel one. As such the volume change of the ProDOT layer will be different as no (or very few) solvent molecules will be incorporated. This could also change the effect of doping on the mechanical properties. Would in this case the polymer stiffen upon oxidation? I do not have an answer but I think the point is relevant as in the end practical devices will have solid state electrolytes.

Apart from that I have a few very minor corrections:

- 1) Pag 3 line 3. "facial" should be "facile"
- 2) pag 3 line 11. "anion insertion in PEDOT" was this PEDOT:PSS?
- 3) Fig 2. Graphs are too small and difficult to read
- 4) pag 15 line 19. Please describe the nanoparticles (dimensions, how they are made)
- 5) pag 20 electrochemistry. Please justify why some experiments are made with 1M LiPF₆ and others with 0.2 M LiTFSi.
- 6) The polymer paperd is very short, is this the best that can be done based on available literature protocols?

Reviewer #3 (Remarks to the Author):

Mei, Zhao and coworkers address a very important issue in electrochromics. Several important issues have to be understood and solved before electrochromic devices based on organic materials will be commercialized on a large scale. Organic electrochromic devices lack sufficient cyclic stability and this study reveals very nicely how mechanical deformation ("breathing") originates from the electron transport and ion intercalation. This deformation affects the properties of the electrochromic coating – a fact that is often ignored by the community. The procedures have been well-defined. The high-quality data is convincingly, well-organized and presented graphically very clearly. This work goes beyond previous studies by others as the authors succeeded here to chemomechanically quantify the "breathing" in poly(3,4-propylenedioxythiophene) – a well-known redox active polymer. Moreover, they succeeded to significantly improve the switching performance of the polymer by about two orders of magnitude by modifying the polymer/electrode interface. A few minor comments: on page 15, the size of the SiNPs and their deposition on the ITO surface is unclear. A better description of the grinding process of the ITO surface would be helpful. The methods section only states "the ITO was grinded by P1200 Starcke silicon carbide sandpaper and then cleaned through the processes mentioned in film processing part." Please add a reference for the preparation of the SiNPs (Stöber method) and mention how these particles were characterized. State the method used to obtain the 3D surface morphology and roughness of the ITO electrode in the caption of figure 7.

Dear Reviewers,

Thank you very much for the very encouraging comments on our work of mechanical breathing in organic electrochromics. We are grateful for the constructive suggestions and questions that you offered to us to improve the quality of the paper. For ease of reference, your original comments are reproduced in **blue**, our response is in **black**, and the resulting changes to the manuscript are in **red**. The following Table R1 provides a quick summary of additional experiments we performed during this revision:

Table R1: Additional experiments to validate the generality of the methodology and results.

Additional experiments	Solvent	Salt	# of cycles
Breathing strain	PC	1M LiPF ₆	100
	1:1 volume ratio EC/DEC	1M LiPF ₆	100
Mechanical properties	PC	1M LiPF ₆	100
	EC/DEC	1M LiPF ₆	1
Cyclic stability	EC/DEC	0.2M LiTFSI	
Characterization of microstructural morphology of SiO ₂ particles by SEM			

More specific point-to-point responses are as follows.

Reviewer 1

Comment 1: “This contribution details the factors that contribute to mechanical delamination of electrochromic polymers from ITO during cycling. The authors employ nanoindentation and modeling of a promising and well-studied polymer, PProDOT. I think the work is generally of interest but I have concerns over data interpretation (comment 9) and the use of SiO₂ (comment 16). Parts of the paper are a bit hard to get through and easily read.”

Response. We appreciate the encouraging comments. We will address the concerns as follows. We have revised the manuscript carefully to make it easier to read.

Comment 2: “Page 3 Paragraph beginning with “In concomitant...” The authors try to generalize behavior of the mechanical properties of electrochemically active polymers. This is dangerous because it is related to the ion type, solvent type, and polymer type. This makes comparisons across refs 28-34 difficult because they have different ions, solvents, and polymers.”

Response. We agree. We noted the significant dependence of the mechanical deformation and properties on the types of polymer/solvent/salt/ion. The following sentence in this paragraph is a reflection of this observation:

“Previous measurements of the mechanical properties of poly(3,4-ethylenedioxythiophene) (PEDOT) using acoustic impedance showed that the shear modulus was sensitive to the doping level²⁸⁻³⁰, temperature^{28,29}, electrolyte^{28,29}, crosslinker^{31,32}, and even film thickness³³.”

The original first sentence in this paragraph “In concomitant...” was not an attempt to make a comparison across references 28-34. It was to state the mechanical breathing strain and a change of mechanical properties induced by the redox reactions in electrochromic polymers. We have revised this sentence as follows:

“The change of the material state in the redox reactions often induces a mechanical breathing strain and a dynamic change of the mechanical properties of the polymers, although there is little consensus in

existing studies on how the mechanical behavior quantitatively evolves over electrochromic processes. Previous measurements of the mechanical properties of poly(3,4-ethylenedioxythiophene) (PEDOT) using acoustic impedance showed that the shear modulus was sensitive to the doping level^{28–30}, temperature^{28,29}, electrolyte^{28,29}, crosslinker^{31,32}, and even film thickness³³.”

Comment 3. “It is my view that the solvent is very important to the mechanical properties, and how the solvent transports in and out of the film, accompanied by an ion’s solvation shell etc.”

Response. Thank you for the insightful comment. We perform an additional measurement of the mechanical properties using 1M LiPF₆ in EC/DEC (volume ratio 1:1) as the electrolyte. Figure R1 shows the comparison using the two solvents (PC vs. EC/DEC). Indeed the mechanical properties of the polymers are dependent on the choice of the electrolyte, but the modulus and hardness decrease by about the same amount after oxidation.

Fig. R1 Mechanical properties of PProDOT film in pristine state (dry and in PC) and at oxidized state in two different electrolytes (LiPF₆ in PC and LiPF₆ in EC/DEC).

Comment 4. “Page 3, “Previous measurement of...” The way it is written, it implies that refs 28-34 all study PEDOT, which is unclear.”

Response. Thank you for the comment. Refs. 28-30 and 32-34 were all about PEDOT.

To make it more concise, we have revised the sentence and adjusted the order of the references as follows:

“Previous measurements of the mechanical properties of poly(3,4-ethylenedioxythiophene) (PEDOT) using acoustic impedance showed that the shear modulus was sensitive to the doping level^{28–30}, temperature^{28,29}, electrolyte^{28,29}, crosslinker^{31,32}, and even film thickness³³.”

- Ispas, A., Peipmann, R., Bund, A. & Efimov, I. On the p-doping of PEDOT layers in various ionic liquids studied by EQCM and acoustic impedance. *Electrochimica Acta* **54**, 4668–4675 (2009).

29. Ispas, A., Peipmann, R., Adolphi, B., Efimov, I. & Bund, A. Electrodeposition of pristine and composite poly(3,4-ethylenedioxythiophene) layers studied by electro-acoustic impedance measurements. *Electrochimica Acta* **56**, 3500–3506 (2011).
30. Schoetz, T. *et al.* Understanding the charge storage mechanism of conductive polymers as hybrid battery-capacitor materials in ionic liquids by in situ atomic force microscopy and electrochemical quartz crystal microbalance studies. *Journal of Materials Chemistry A* **6**, 17787–17799 (2018).
31. Ouyang, L. *et al.* Poly[3,4-ethylene dioxythiophene (EDOT)- co -1,3,5-tri[2-(3,4-ethylene dioxythienyl)]-benzene (EPh)] copolymers (PEDOT- co -EPh): optical, electrochemical and mechanical properties. *Journal of Materials Chemistry B* **3**, 5010–5020 (2015).
32. Qu, J., Ouyang, L., Kuo, C. & Martin, D. C. Stiffness, strength and adhesion characterization of electrochemically deposited conjugated polymer films. *Acta Biomaterialia* **31**, 114–121 (2016).
33. Lyutov, V., Gruia, V., Efimov, I., Bund, A. & Tsakova, V. An acoustic impedance study of PEDOT layers obtained in aqueous solution. *Electrochimica Acta* **190**, 285–293 (2016).
34. Wang, S., Li, F., Easley, A. D. & Lutkenhaus, J. L. Real-time insight into the doping mechanism of redox-active organic radical polymers. *Nature Materials* **18**, 69 (2019).

Comment 5. “Page 8. The equation for volumetric strain assumes a constant area, but the video clearly shows changes in area. Authors should discuss.”

Response. The change in area shown in the video only occurs in the delaminated regimes where the polymer film lost contact with the substrate after ~160 cycles. For the regimes well bonded with the substrate (where electrochromic switching remains functioning in the video), the in-plane deformation (areal change) of the film is constrained by the hard substrate. When we measure the change of the film thickness at the first cycle, the film is well-adhered to the substrate. We also excluded the data from the indent locations which are within ~50um from the edge to avoid the interference of the film delamination. We added a discussion about this precaution.

“With the two methods described above, we measure the change of thicknesses of the film at the same locations in the pristine and oxidized state in the first cycle. The nanoindentation sites are chosen ~50 um away from the edge to avoid possible interference of film delamination from the substrate. As seen in Fig. 2b, the film surface is clearly elevated upon oxidation indicating an increase of the film thickness. For each measured location, we compare the thicknesses of the film before and after oxidation, (h_0, h) . Since the in-plane deformation of the film is bounded by the hard substrate, the volumetric strain is calculated by $\varepsilon_V = (h - h_0) / h_0$.”

Comment 6. ““This overall volumetric strain gives a roughly 10% linear strain for a homogeneous and isotropic material.” This sentence is confusing given comment #6.”

Response. The volumetric strain is calculated by $\varepsilon_V = (1 + \varepsilon)^3 - 1$ assuming an isotropic deformation in every direction. When $\varepsilon \ll 1$ (small deformation), $\varepsilon \cong \varepsilon_V / 3$. This linear strain is needed for the finite element modeling.

Comment 7. “Figure 3 and caption should be understandable as a self-contained unit. Panel (a) – is it dry or in PC? Panel (b) – are “In PC” samples reduced or oxidized. Panel (b) are “Oxidized” samples dry or in PC?”

Response. Thank you for the suggestion. Figure 3 has now been replaced by a new figure with additional experimental results. Please see the Response to Comment 9.

Comment 8. ““We perform nanoindentation to measure the elastic modulus and hardness of the PProDOT film in the pristine and oxidized states using the continuous stiffness measurement 1 (CSM).” Dry or in PC?”

Response. Thank you for the comment. We revise the sentence as follows to make it clearer:

“We perform nanoindentation to measure the elastic modulus and hardness of the PProDOT film in the pristine state (dry and in PC), reduced state (in electrolyte), and oxidized state (in electrolyte) using the continuous stiffness measurement (CSM).”

Comment 9. “I have concerns with Figure 3. Many electrochemically active polymers need conditioning for electrolyte penetration. I am not sure that the authors considered this. I think that the “In PC” and “Oxidized” measurements are just a result of the electrode not being properly conditioned. To resolve this the authors should present data on “Uncycled In PC”, “Conditioned and Reduced”, and “Conditioned and Oxidized”. Then it will be a clearer result. Without addressing conditioning, I fear their discussions on pages 9-10 are misguided.”

Response. We fully agree with the concern about electrochemical conditioning. We conduct the same experiments – breathing strain measurement and mechanical properties measurement – upon the first 100 cycles. The goal is to evaluate the impact of the electrochemical conditioning, as suggested by the reviewer. The new Figure 3 is copied as follows. Interestingly, the mechanical properties of the polymer in the 1st and the 100th cycle are not very different. Both the modulus and hardness at the reduced state are more than a factor of two of that at the oxidized state.

Fig. 3 Mechanical properties of PProDOT film. a Upper panel: The load-displacement curves of indentation on the pristine and oxidized PProDOT films. Lower panel: Modulus and hardness of the pristine and oxidized PProDOT as a function of the indentation depth. b Modulus (upper panel) and hardness (lower panel) of PProDOT in the pristine and dry state, the pristine state in PC, the oxidized state in electrolyte after the 1st cycle, the reduced state in electrolyte after 100 cycles, and the oxidized state in electrolyte after 100 cycles.

In addition, this comment motivated us to evaluate two important factors: (1) the mechanical breathing strain of the film after conditioning, and (2) the solvent effect. Here we measure the breathing strain *in-situ* using the electrolytes of LiPF₆ in PC and LiPF₆ in EC/DEC (1:1 volume ratio) for the first 100 cycles.

The experimental procedure using LiPF_6 in PC is described as follows. We hold the voltage at 1 V for 10s and then at -0.2V for 10s for the first 50 cycles. We then hold the voltage at 1 V for 5s and at -0.2V for 5s for the 51-100 cycles. The switch time between 1V and -0.2V is 1s. Using nanoindenter, we hold the tip on the surface of the film with a very small load ($\sim 3\mu\text{N}$) during electrochemical cycles. The raw displacement is recorded to calculate the breathing strain. The experimental procedure using LiPF_6 in EC/DEC is the same, except that the hold time is 5s throughout over 100 cycles. The following Fig. R2 shows the breathing strain – 15~20% using LiPF_6 in PC and $\sim 10\%$ using LiPF_6 in EC/DEC, throughout the cycles.

Some interesting observations in the preliminary results: (1) We do not observe the first-cycle effect. The breathing strain in the first cycle and in the following cycles are about the same. (2) The value of the breathing strain depends on the choice of the solvent but is close for the two electrolytes we have tested. (3) Using both electrolytes, the breathing strain fades away after ~ 20 cycles, but upon the resume of the electrochemical reaction, the mechanical strain is reactivated. (4) The breathing strain is persistent within the third block of cycles.

From the experiments we can conclude that the mechanical breathing strain and the mechanical properties are not very sensitive to the electrochemical conditioning as the reviewer questioned. However, we must admit that we have not reached a full understanding, for instance, what are the timescales of ion transport and molecular relaxation relative to the holding time in experiments? What is the effect of the molarity of Li salt? How about other types of electrolytes, for instance, gel electrolyte? What is the time-dependent mechanical properties in the transient states of the redox reactions? This paper will unlikely address all these outstanding questions. We decide not to include the preliminary results in Fig. R2 in the current manuscript because the dynamic behavior is not fully understood. We hope that the reviewer shares the same opinion with us – this study potentially opens a wide space of studies pertinent to the mechanical reliability of organic electrochromics which warrants further systematic studies.

Fig. R2 The mechanical breathing strain of the film during the first 100 cycles, **a** using 1M LiPF₆ in PC, and **b** using 1M LiPF₆ in EC/DEC.

Comment 10. “Figure 4 is hard to interpret for the non-expert. The ITO and polymer film should be labeled. The crack propagation or delamination should be indicated, perhaps with an arrow.”

Response. Thank you for the suggestion. We updated Figure 4 by removing the reference frames of the film. Now the figure should have become easier to understand. We also added the labels here for the reviewer’s reference.

Fig. 4 Contour plots of the shear stress τ_{xy} in PProDOT at the oxidized state, τ_{xy} at the reduced state, and the normal stress σ_y at the reduced state after the 1st, 4th, and 8th cycles, respectively. Deformation is drawn in the actual scale.

Comment 11. “Page 15 “In short, the general guideline is to use small-size, stiff, and soft film electrode, and tough interfacial adhesion.” How can an electrode be both “stiff” and “soft”? That seems contradictory. Same issue in the conclusion section.”

Response. In the language of solid mechanics, “soft” refers to the plastic property of materials when the yield strength is low, while “stiff” refers to the elastic property of materials and the Young’s modulus is high. In principle, the “softness” and “stiffness” may be tuned separately, that is, one material may behave liquid-like when it flows but behave solid-like in the elastic stage. However, we are not aware of any polymers showing “softness” and “stiffness” at the same time for now. This is probably very challenging in material design, and that is the reason that we focus on improving the interfacial strength which is much easier to achieve.

To clarify this, we have revised the sentence:

“In short, the general guideline is to use small-size, stiff (**high modulus**), and soft (**low yield strength**) film electrode, and tough interfacial adhesion.”

We have removed this repeated description in conclusion to avoid further confusion.

Comment 12. “Page 14. The energy release rate G depends on modulus and yield strength, but these vary between oxidized and reduced states. How did the authors account for this? Or what assumptions went into the calculation?”

Response. Thank you for the question. The computational modeling was not intended to simply replicate the interfacial failure in the long-term cycle of the polymer as we observed in experiments. We set constant values of modulus and yield strength in the numerical modeling. To avoid the change of results by using different sets of material parameters, we construct the phase diagram in terms of the normalized quantities – the phase diagram will not be altered as long as the dimensionless parameters remain the same in the modeling.

Comment 13. ““For a given selection of the polymeric film, the mechanical properties are about fixed.” Confusing. What is really meant here?”

Response. For the family of the PProDOT polymers, the modulus E and shear yield strength τ_c can be tuned through molecular design and film processing, but the resulting change to the dimensionless parameter $\frac{\Gamma E}{Z\tau_c^2 h_0}$ is not very significant. The original description was to motivate the design to be focused on the modifying the interfacial strength. To avoid this confusion, we have removed this sentence.

The description is as follows now:

“For the fabrication and device performance, the thickness of the film electrode is often chosen to maximize the optical contrast between the two redox states. Among the rules offered by the phase diagram, the interfacial toughening by physical or chemical modification seems most practical.”

Comment 14. “Please do not use SiO₂ as (SiNP) because in the silicon anode community that term means “Si” pure nanoparticles – not SiO₂, so it will confuse audience.”

Response. Thank you very much for the suggestion. We have made this correction throughout the paper.

Comment 15. “Figure 6. Do all electrodes have the same thicknesses?”

Response. Yes, all the electrodes were prepared under the same conditions, i.e., ITO cleaned under same condition, the same batch of solution (40 mg/mL), and spin-coated of the same speed (1500 rpm) as we described in the Methods section. The electrodes are of the similar thicknesses.

Comment 16. “Why chose SiO₂? It is insulating and not an obvious choice. I am surprised that an SiO₂ interface still allows for electrochemical activity between the ITO and the polymer.”

Response. SiO₂ nanoparticles are cheaper compared to other choices, for example, carbon nanotube and silver nanoparticles. Another advantage of SiO₂ is its ability to allow air-processability. The nanoparticles described here would form self-assembled and well-ordered pattern, as seen in the AFM image in Figure 7 and also the SEM image in Supplementary Fig. 6 (response to the comment 4 of Reviewer 2). The porous nature of the structure allows penetration of the polymer into the SiO₂ layer, therefore the polymer remains in contact with the ITO current collector. Although SiO₂ is insulating in its bulk form, the SiO₂ nanolayer used here would permit conductivity.

Reviewer 2

Comment 1. “The paper describes a very informative study on one of the most relevant topics in organic electrochromic devices: the lack of stability after repeated cycling due to delamination. OEC materials have advantages over the corresponding inorganic counterparts such as WO₃, yet the relatively poor stability they feature because of intrinsic chemical sensitivity to oxygen and water and mechanical instability in multilayer devices so far hampered the commercial valorisation of results.

If careful (and expensive) encapsulation can dramatically improve chemical sensitivity, the problem of delamination and cracking connected with the large volume variation upon oxidation/reduction is more difficult to tackle.

I particularly appreciate the results here described because the quantitative evaluation of the extension of volume variation is professionally carried out leading to values that are fully believable, and very useful for the community.

Also, to the best of my knowledge this is the first time that the changes in the mechanical properties upon doping are directly measured, and with very surprising results. I did not expect the ProDOT to soften upon chemical doping, yet the data are believable and the interpretation is sound.

The finite element analysis of the crack propagation is also very relevant, particularly as useful guidelines can be obtained in order to design improved working electrodes (essentially by improving adhesion).”

Response. Thank you so much for the many encouraging comments! We particularly appreciate the insight of the reviewer about the intrinsic mechanical instability of the multilayer device that is resulted from the mechanical breathing strain in the redox active polymers – we cannot agree more on this point.

Comment 2. “I have only one relevant concern about the paper: generality.

Essentially all aspects here described are bound to be solvent and electrolyte dependent. Indeed, the stability of ProDOT upon cycling was found to be very good already quite a few years ago, when working with acetonitrile as the solvent. (10.1002/adma.200300376). Multiple reports highlight both the high electrochromic contrast and high stability of such a polymer.

I do not doubt the data described here but it looks like the particular solvent here employed was selected to highlight the degradation by delamination phenomenon. I was particularly surprised by the data shown in figure 6c of such a severe damage after just 140 cycles.

I strongly suggest to repeat the stability upon cycling test in at least another solvent (acetonitrile could be a good choice) and to use an electrolyte more relevant for devices (LiTFSI could be good).”

Response. Thank you for the question about the generality. As seen in the summary of the additional experiments we performed in this revision (Table R1), we measured the mechanical breathing strain, mechanical properties, and cyclic stability of PProDOT by changing the solvent (PC and EC/DEC) and salt (LiPF₆ and LiTFSI) and testing upon electrochemical conditioning. Please see the Response to the comment 3 and comment 9 of Reviewer 1 for detailed discussions of the results.

Thank you for the suggestion on acetonitrile. Acetonitrile is very volatile which gives some difficulty in the open-cell measurement – the solvent quickly evaporates during the measurements of the breathing strain, mechanical properties, and also during the stability test which takes several days. Here we took the liberty to choose EC/DEC (volume ratio 1:1) as another solvent. All other procedures for the stability test remain the same as previous experiments.

The results using EC:DEC are shown in Figure R3. The bare ITO, roughened ITO, and SiO₂ NP coated ITO reach the cycles (current density > 0.08 mA cm⁻²) of 1800, 2500, 2750, respectively. The overall performance is certainly not optimal, but again demonstrates the mitigation of mechanical instability in electrochromic films by interfacial toughening. We observe significant thinning of the

polymer film at prolonged cycles for all three types of electrodes, which might be due to the film dissolving into the solvent. Also, since DEC is volatile the electrolyte becomes more viscous over cycles. These factors are detrimental to the performance of the electrodes.

On the other side, comparing the two solvents PC and EC:DEC, the cyclic performance of bare ITO electrode and roughened ITO electrode in EC:DEC largely improves. As shown in Figure R2, the breathing strain of the film is smaller in EC/DEC, which reduces the probability of the film delamination over cycles. In addition, the oxidation onset is around 0.2 V for all three sets of electrodes, which indicates that the interfacial modification does not alter the electrochemistry of the PProDOT film. Comparing to the results in PC, the current density using EC:DEC is slightly smaller, and the oxidation onset is lower.

Together with other additional experiments summarized in Table R1, the stability test using EC:DEC serves the purpose of validating the generality of the methodology and the findings. The cyclic results are not ideal and require much more optimization in our future work. If the reviewer permits, we would not attempt to include this new result in the current paper. We will pursue and publish a better understanding/result in another work.

Fig R3. Interfacial modification of electrochromic electrodes in 0.2M LiTFSI in EC/DEC. **a** Cyclic lifetime of PProDOT electrodes. **b-d** show the cyclic voltammometry response of PProDOT film on bare ITO, roughened ITO, and SiO₂ NP-treated ITO, respectively.

Comment 3. “Again on the same topic, in solid state devices the liquid electrolyte is generally replaced by a gel one. As such the volume change of the ProDOT layer will be different as no (or very few) solvent molecules will be incorporated. This could also change the effect of doping on the mechanical properties. Would in this case the polymer stiffen upon oxidation? I do not have an answer but I think the point is relevant as in the end practical devices will have solid state electrolytes.”

Response. Gel electrolyte is definitely more relevant in solid-state devices. For the question that whether the polymer softens upon oxidation when the liquid solvent is replaced by a solid one, we unfortunately

do not have a clean answer either. We can place some speculations as we discussed in the manuscript on how we think that the ion intercalation in general weakens the inter-molecular interactions of the polymers, but clearly the mechanical behavior would depend on the particular choice of the solvent/salt/ion. The *in-situ* measurement of all-solid multilayer device requires another design of the apparatus which is not doable in the current experimental setup. This is our long-term goal and we are actively pursuing the understanding of the mechanical behavior/reliability in different settings of the electrochromic system.

Comment 4. “Apart from that I have a few very minor corrections:

1) Pag 3 line 3. “facial” should be “facile”

Response. Thank you for the careful read, we have made this correction.

2) pag 3 line 11. “anion insertion in PEDOT” was this PEDOT:PSS?

Response. The cited paper is about PEDOT in various ionic liquids.

3) Fig 2. Graphs are too small and difficult to read

Response. Thank you, we have revised this figure and copied as follows.

4) pag 15 line 19. Please describe the nanoparticles (dimensions, how they are made)

Response. We now describe the silica nanoparticles in more detail as copied below. The syntheses procedure is described in the *Methods* section under *Surface modification of ITO*.

The bare ITO has the finest surface with a root mean square height of only 5.51 nm, followed by SiO_2 NP treated ITO surface (21.9 nm). The nanoparticles (diameter of $\sim 200\text{nm}$) self-assemble into a well-packed hierarchy nanostructure, as shown in the 3D AFM image in Fig. 7(d) and in the SEM image in Supplementary Fig. 6. Nanoscale interparticle gaps (white arrow in Supplementary Fig. 6) introduces high-density mechanical interlock between the polymer film and the substrate, which improves the electrochemical performance. Note that the mud cracks (red arrows) in Supplementary Fig. 6 are formed by the electron-wind forces at high magnification in SEM.

Supplementary Figure 6. Scanning electron microscopy images of SiO₂ nanoparticles deposited on ITO-glass substrate. White arrow indicates interparticle gaps. Red arrows indicate mud cracks induced by electron-wind forces during SEM imaging.

5) pag 20 electrochemistry. Please justify why some experiments are made with 1M LiPF₆ and others with 0.2 M LiTFSi.

Response. We chose 1M LiPF₆ in the mechanical strain and mechanical properties measurements, while we chose 0.2M LiTFSI (in Dr. Mei's lab) for the cyclic stability test. We did not attempt to make a comparison between the mechanical and the electrochemical tests, however, we could certainly plan better on the experiments at the earlier stage. Now with the new experiments performed using both LiPF₆ and LiTFSI for both mechanical measurements and electrochemical stability tests (Table R1), there is more consistency in the experiments and conclusions now.

6) The polymer paperd is very short, is this the best that can be done based on avaiable literature protocols?"

Response. The polymer film shown in the video was specially prepared (~1mm × 1.5mm) so we can observe it under the optical microscope. For the electrochemical cycles, the film was spin-coated on the ITO-glass substrate and the polymer size is 8mm × 40mm. This size is more commonly used in literature^{1,2}.

1. He, Jiazhi, Liyan You, and Jianguo Mei. "Self-bleaching behaviors in black-to-transmissive electrochromic polymer thin films." *ACS applied materials & interfaces* 9.39 (2017): 34122-34130.
2. De Keersmaecker, Michel, et al. "All Polymer Solution Processed Electrochromic Devices: A Future without Indium Tin Oxide?." *ACS applied materials & interfaces* 10.37 (2018): 31568-31579.

Reviewer 3

Comment 1. “Mei, Zhao and coworkers address a very important issue in electrochromics. Several important issues have to be understood and solved before electrochromic devices based on organic materials will be commercialized on a large scale. Organic electrochromic devices lack sufficient cyclic stability and this study reveals very nicely how mechanical deformation (“breathing”) originates from the electron transport and ion intercalation. This deformation affects the properties of the electrochromic coating – a fact that is often ignored by the community. The procedures have been well-defined. The high-quality data is convincingly, well-organized and presented graphically very clearly. This work goes beyond previous studies by others as the authors succeeded here to chemomechanically quantify the “breathing” in poly(3,4-propylenedioxythiophene) – a well-known redox active polymer. Moreover, they succeeded to significantly improve the switching performance of the polymer by about two orders of magnitude by modifying the polymer/electrode interface.”

Response. Thank you very much for the very encouraging comments!

Comment 2. “A few minor comments: on page 15, the size of the SiNPs and their deposition on the ITO surface is unclear.”

Response. Thank you for the question. Please refer to the Response to Comment 4 of Reviewer 2.

Comment 3. “A better description of the grinding process of the ITO surface would be helpful.”

Response. Thank you for the suggestion. Now we add more details about the grinding process in the *Methods* section.

“In the first method, the ITO was grinded by P1200 Starcke silicon carbide sandpaper. **Very gentle force is applied in two orthogonal directions in sequence to generate visible clouds on ITO surface. The grinded ITO was then cleaned through the processes mentioned in film processing part.**”

Comment 4. “The methods section only states “the ITO was grinded by P1200 Starcke silicon carbide sandpaper and then cleaned through the processes mentioned in film processing part.” Please add a reference for the preparation of the SiNPs (Stöber method) and mention how these particles were characterized.

Response. Thank you for the comment. We followed the procedure described by Werner Stöber and coauthors (Werner Stöber, Arthur Fink, and Ernst Bohn. "Controlled growth of monodisperse silica spheres in the micron size range." *J. Colloid. Interf. Sci.* 26.1 (1968): 62-69.). We added the reference to the manuscript. For the characterization of the nanoparticles, please see Response to Comment 4(4) of Reviewer 2.

Comment 5. State the method used to the obtained 3D surface morphology and roughness of the ITO electrode in the caption of figure 7.”

Response. Thank you for the suggestion. The 3D surface morphology was obtained via atomic force microscope (tapping mode). The surface roughness was calculated via an online tool at <https://www.profilmonline.com/>. We revise the caption of Figure 7 as follows:

“Fig. 7 3D surface morphology **by AFM** and the roughness of the ITO surface. a Bare ITO. b A flat region in roughened ITO. c A scratched region in roughened ITO. d SiO₂ NP-treated ITO. Sq denotes the root mean square height roughness.”

We also add a description about the calculation of the surface roughness *Methods* section:

“The AFM (Veeco Multimode) data for comparison is processed via an online tool at <https://www.profilmonline.com/>.”

REVIEWERS' COMMENTS:

Reviewer #1 (Remarks to the Author):

The authors have addressed all but one of my comments. They authors made a very nice Figure 4 in their response (with better labels), but that Figure 4 did not make it to the main text. Could the authors please update Figure 4.

Reviewer #2 (Remarks to the Author):

The authors made an excellent job on sizeably improving the paper. I have no further comments.

Reviewer #3 (Remarks to the Author):

The authors have significantly revised their manuscript and addressed the questions. In my opinion, this important study can be published. No further revision is required.

Reviewer #1 (Remarks to the Author):

Comment: The authors have addressed all but one of my comments. They authors made a very nice Figure 4 in their response (with better labels), but that Figure 4 did not make it to the main text. Could the authors please update Figure 4.

Response: Thank you for your suggestion. We have revised Figure 4 in the main text with proper labels.

Fig. 4 Contour plots of the shear stress τ_{xy} in PProDOT at the oxidized state, τ_{xy} at the reduced state, and the normal stress σ_y at the reduced state after the 1st, 4th, and 8th cycles, respectively. Deformation is drawn in the actual scale.